# Aligning Latent Spaces with Flow Priors

## Abstract

This paper presents a novel framework for aligning learnable latent spaces to arbitrary sample-defined prior distributions by leveraging flow-matching models as expressive priors. Our method first pretrains a flow-matching model on prior features to capture their conditional interpolation dynamics. The fixed flow model subsequently regularizes the latent space via an alignment loss, which reformulates the flow-matching objective by treating the learnable latents as candidate endpoints. We establish a score-space foundation for this objective: under the population-optimal conditional flow-matching field, the pointwise alignment loss is exactly a multiscale relative Fisher energy between candidate-induced Gaussian probes and Gaussian-smoothed prior marginals. For isotropic Gaussian priors, this energy is an affine transformation of the exact negative log-likelihood; at the aggregate level, its expectation decomposes into a proper multiscale Fisher discrepancy and an endpoint-recoverability term. These results provide a principled basis for using the alignment loss as a computationally tractable prior-likelihood surrogate. Notably, the proposed method eliminates expensive likelihood evaluations and avoids ODE solving during downstream optimization. In controlled experiments, the alignment-loss landscape closely approximates the negative-log-likelihood landscape of diverse two-dimensional priors. We further validate the effectiveness of our approach by regularizing autoencoder latent spaces with diverse visual and textual priors in large-scale ImageNet image generation, accompanied by detailed discussions and ablation studies. With both theoretical and empirical validation, our framework paves a new way for flexible latent-space alignment.

## 1 Introduction

Latent models like autoencoders (AEs) are a cornerstone of modern machine learning (Hinton & Salakhutdinov, 2006; Baldi, 2012; Li et al., 2023; Chen & Guo, 2023; Mienye & Swart, 2025). These models typically map high-dimensional observations to a lower-dimensional latent space, aiming to capture salient features and dependencies (Liou et al., 2014; Meng et al., 2017). A highly desirable property of latent models is that the latent space should have structural properties, such as being close to a predefined prior distribution (Rifai et al., 2011; Kingma & Welling, 2014; Yao et al., 2025; Chen et al., 2024). Such structure can incorporate domain-specific prior knowledge (Khemakhem et al., 2020; Raissi et al., 2019), enhance the interpretability of the latent space (Higgins et al., 2017; Chen et al., 2016; Kim & Mnih, 2018), and facilitate latent space generation (Rombach et al., 2022; Li et al., 2024a; Leng et al., 2025; Wen et al., 2025; Yu et al., 2025). For example, in latent generative models, encouraging autoencoder latents to follow a structured prior enables high-quality synthesis.

Traditional approaches to enforcing distributional conformity often involve minimizing divergences like the Kullback–Leibler (KL) divergence (Kingma & Welling, 2014; Rombach et al., 2022). However, KL can be restrictive, particularly when the prior is only implicitly defined by samples. In latent generative modeling, the latent space is usually regularized with known prior distributions, such as the Gaussian distribution for Variational Autoencoders (VAEs) (Kingma & Welling, 2014), and the categorical distribution for Vector

LLM Usage Declaration: This paper makes use of large language models (LLMs) exclusively for non-substantive assistance, including grammar correction, text formatting, LaTeX syntax support, and proofreading. All research ideas, mathematical formulations, experimental design, analysis, and conclusions are the original work of the authors.

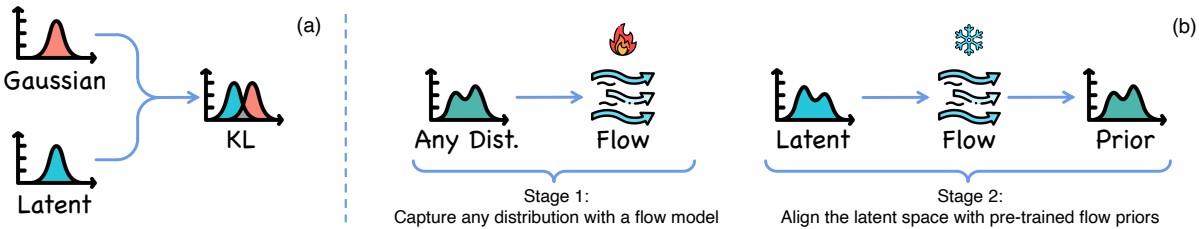

Figure 1: (a) Conventional alignment uses explicit parametric priors, such as Gaussian or categorical distributions, together with objectives such as KL divergence or cross-entropy. (b) Our method first captures an arbitrary sample-defined prior with a flow-matching model and then uses the frozen flow prior to align the learnable latent space.

Quantized VAEs (VQ-VAEs) and VQ-GANs (Van Den Oord et al., 2017; Esser et al., 2021). Recent works (Qiu et al., 2025; Li et al., 2024b;c; Chen et al., 2025b; Yao et al., 2025; Chen et al., 2024) have proposed to use pre-trained feature extractors as samples that define the prior over latents and directly optimize latent distances. These methods are effective, but they require per-sample features during autoencoder training and can introduce substantial computational cost.

Recent advances in flow-matching (FM) generative models (Lipman et al., 2023; Liu et al., 2023) offer a promising avenue to capture complex prior distributions. In this work, we address the question: *Can we efficiently align a learnable latent space to an arbitrary sample-defined prior distribution using a pre-trained FM model as a prior?* We answer this question affirmatively by proposing a novel framework that leverages a pre-trained FM model to define a computationally tractable alignment loss, which effectively guides the latents toward the prior distribution.

Our proposed approach unfolds in a two-stage process, as illustrated in Fig. 1. The first stage involves pretraining an FM model on the desired prior features, allowing it to learn the conditional velocity field associated with Gaussian-to-prior interpolation paths. Once this flow model captures the prior dynamics, its parameters are fixed. In the second stage, the frozen flow model serves as a prior to regularize a learnable latent space, for instance, the output of the encoder in an AE. This regularization is achieved by minimizing an alignment loss that adapts the standard FM objective by treating the learnable latents as candidate endpoints. The resulting pipeline guides the latent space toward the desired prior structure without requiring direct comparison to prior samples or expensive likelihood evaluations of the flow model.

We theoretically justify the method by characterizing the alignment loss as a multiscale prior energy. Under the population-optimal conditional FM field, the pointwise loss is exactly a time-averaged, weighted relative Fisher divergence between the Gaussian probe induced by a candidate latent and the Gaussian-smoothed prior marginal at each interpolation time. This identity provides a direct score-space measure of how well a candidate endpoint agrees with the prior dynamics learned by the flow model. For isotropic Gaussian priors, the resulting energy is an affine transformation of the exact negative log-likelihood (NLL); for general priors, it extends this calibration through a multiscale Fisher geometry. At the aggregate-distribution level, the expected loss decomposes into a proper multiscale Fisher discrepancy and an endpoint-recoverability term, explaining how the objective simultaneously transfers prior structure and promotes a compact, modelable latent space.

Our framework offers three key advantages. First, our approach enables alignment to **arbitrary sample-defined prior distributions**, including continuous feature distributions and discrete codebook-embedding distributions, without requiring an explicit parametric density. Second, the alignment loss acts as a **theoretically characterized likelihood surrogate**: it exactly recovers isotropic-Gaussian NLL up to scale and offset, while its general form measures multiscale score agreement with the prior. This provides a principled alternative to heuristic metrics such as cosine similarity used in per-sample feature matching (Chen et al., 2025b; Yao et al., 2025; Chen et al., 2024). Third, our framework is **computationally lightweight**, requiring only a single forward pass through the pre-trained flow model during training and bypassing expensive

adversarial prior matching (Makhzani et al., 2015), likelihood evaluation, ODE solving, and per-sample feature extraction (Qiu et al., 2025; Li et al., 2024b;c).

We empirically validate the efficacy of the proposed alignment strategy through a series of experiments. We begin with controlled two-dimensional priors and demonstrate that the alignment-loss landscape closely approximates the NLL landscape, while minimizing the loss consistently moves learnable variables toward high-likelihood regions. We then demonstrate the scalability of the approach through large-scale latent image generation experiments on ImageNet (Deng et al., 2009) with diverse visual and textual prior distributions. Detailed discussions and ablation studies further characterize the reconstruction–generation trade-off and the effect of the alignment strength.

We believe this method offers a powerful and flexible tool for incorporating rich distributional priors into latent models. Our work paves the way for more flexible and powerful structured representation learning, and we anticipate its application and extension in domains requiring distributional control over latent spaces.

## 2 Related Work

### 2.1 Flow-based Generative Models and Flow Priors

Flow-based generative models provide a way to represent complex distributions through transport maps. Early normalizing flows, including NICE, Real NVP, and Glow, built exactly invertible transformations with tractable change-of-variable likelihoods (Dinh et al., 2014; 2017; Kingma & Dhariwal, 2018). Continuous normalizing flows extend this view to ODE-defined transformations, but likelihood evaluation requires solving an ODE and computing Jacobian traces (Chen et al., 2018; Grathwohl et al., 2019). Flow Matching (FM) and related stochastic-interpolation or rectified-flow objectives train continuous-time vector fields by regressing velocities along probability paths, avoiding likelihood computation during model fitting (Lipman et al., 2023; Albergo & Vanden-Eijnden, 2023; Liu et al., 2023; Neklyudov et al., 2023; Heitz et al., 2023; Tong et al., 2023). Recent large-scale flow models further show that such objectives can scale to high-resolution image and video generation (Esser et al., 2024; Labs, 2024; Chen et al., 2025c; Zhai et al., 2025; Zhao et al., 2024; Shin et al., 2025). Beyond generation, pre-trained flow priors have also been used as regularizers for inverse problems (Zhang et al., 2024). Our work uses a pre-trained FM model differently: the flow is frozen after learning a reference feature distribution, and its velocity field is reused as a tractable latent-space regularizer.

### 2.2 Latent Priors in Autoencoders

Latent autoencoders usually impose simple structural assumptions on their latent spaces. VAEs regularize the posterior toward a Gaussian prior through a KL term (Kingma & Welling, 2014), while VQ-VAE and VQ-GAN restrict latents to a discrete codebook (Van Den Oord et al., 2017; Esser et al., 2021). More expressive VAE priors have been explored through hierarchical latent variables and autoregressive or flow-based posterior families (Sønderby et al., 2016; Vahdat & Kautz, 2020; Kingma et al., 2016). These designs improve flexibility, but the prior is still largely specified by a parametric form or by the architecture of the latent variable model. In contrast, recent tokenizer work increasingly treats the latent distribution itself as a central design choice for downstream generative modeling. SoftVQ-VAE introduces a continuous tokenizer based on soft categorical posteriors, achieving high compression while maintaining a semantically rich latent space (Chen et al., 2024). DMVAE explicitly constrains the aggregate latent distribution to match an arbitrary reference distribution, including self-supervised features, diffusion noise states, and other empirically defined priors (Ye et al., 2025). Our method is aligned with this distribution-level perspective, but it uses a frozen flow prior to define a lightweight optimization signal rather than jointly training score models for aggregate-posterior matching.

### 2.3 Representation-aligned Tokenizers and Generative Latents

Another active direction improves generative latents by borrowing structure from pre-trained representation encoders. Several methods regularize autoencoder latents with features from DINO, MAE, text-image encoders, or related semantic teachers (Qiu et al., 2025; Li et al., 2024b;c; Chen et al., 2025b; Yao et al., 2025; Chen et al., 2024; Kim et al., 2025; Zha et al., 2024). These approaches show that semantically structured

latents can improve the diffusability or autoregressive modelability of the latent space, but many of them rely on per-sample alignment losses between each image latent and a corresponding teacher feature. More recent work pushes this idea further. AlignTok starts from a pre-trained visual foundation encoder and adapts it into a continuous tokenizer for diffusion models, emphasizing that semantically grounded latent spaces improve convergence and generation (Chen et al., 2025a). Representation Autoencoders (RAEs) replace conventional VAE encoders with frozen representation encoders such as DINO, SigLIP, or MAE, paired with trained decoders, showing that representation latents can support strong reconstruction and diffusion training when the downstream architecture is adapted (Zheng et al., 2025). These methods directly modify the tokenizer or downstream generator. Our work instead asks whether a learnable latent space can be regularized toward a sample-defined reference distribution through a separate pre-trained flow prior.

## 2.4 Representation Alignment during Generator Training

Representation alignment has also been applied inside the generative model rather than only at the tokenizer level. REPA aligns intermediate diffusion-transformer features with representations from non-generative encoders to accelerate training (Yu et al., 2025), and REPA-E extends this idea to end-to-end tuning of VAE-based latent diffusion transformers (Leng et al., 2025). HASTE observes that persistent representation alignment can eventually constrain the generative student, and proposes a staged schedule that applies holistic teacher alignment early and terminates it later (Wang et al., 2025). These methods use representation teachers to guide the dynamics or internal features of a diffusion model. Our setting is complementary: the downstream generator need not be aligned to a teacher during its own training, because the prior information is distilled beforehand into a frozen FM model and then used as a regularizer for the autoencoder latents.

# 3 Preliminaries

## 3.1 Flow Matching

We consider an ODE defined by a time-dependent velocity field $\boldsymbol{u}(\boldsymbol{x}_t, t)$:

$$\frac{\mathrm{d}\boldsymbol{x}_t}{\mathrm{d}t} = \boldsymbol{u}(\boldsymbol{x}_t, t), \qquad \boldsymbol{x}_0 \sim p_{\mathrm{init}}. \tag{1}$$

Here, $p_{\mathrm{init}}$ is a simple base distribution, such as $\mathcal{N}(\boldsymbol{0}, \boldsymbol{I})$. Assuming the field is Lipschitz in its spatial argument and continuous in time, the ODE defines a deterministic flow map.

Since the target velocity field is unknown, we approximate it with a neural network $\boldsymbol{v}_\theta(\boldsymbol{x}_t, t)$:

$$\frac{\mathrm{d}\boldsymbol{x}_t}{\mathrm{d}t} = \boldsymbol{v}_\theta(\boldsymbol{x}_t, t), \qquad \boldsymbol{x}_0 \sim p_{\mathrm{init}}. \tag{2}$$

Flow matching (Lipman et al., 2023; Liu et al., 2023) trains $\boldsymbol{v}_\theta$ by regressing conditional velocities along a prescribed probability path. Let $\boldsymbol{X}_0 \sim p_{\mathrm{init}}$ and $\boldsymbol{X}_1 \sim p_{\mathrm{prior}}$ be independent. A common choice is the linear interpolation

$$\boldsymbol{X}_t = (1 - t)\boldsymbol{X}_0 + t\boldsymbol{X}_1, \qquad t \in [0, 1], \tag{3}$$

whose sample-wise conditional velocity is $\boldsymbol{X}_1 - \boldsymbol{X}_0$. The corresponding flow-matching objective is

$$\mathcal{L}_{\mathrm{FM}}(\theta) = \mathbb{E}_{t, \boldsymbol{X}_0, \boldsymbol{X}_1} \left[ \|\boldsymbol{v}_\theta((1 - t)\boldsymbol{X}_0 + t\boldsymbol{X}_1, t) - (\boldsymbol{X}_1 - \boldsymbol{X}_0)\|^2 \right], \tag{4}$$

where $t \sim \mathcal{U}[0, 1]$ independently of $(\boldsymbol{X}_0, \boldsymbol{X}_1)$.

Under squared loss, the population-optimal deterministic regressor is the conditional mean velocity

$$\boldsymbol{v}_P^\star(\boldsymbol{x}, t) = \mathbb{E}[\boldsymbol{X}_1 - \boldsymbol{X}_0 \mid \boldsymbol{X}_t = \boldsymbol{x}], \tag{5}$$

where $P = p_{\mathrm{prior}}$. For any measurable field $\boldsymbol{v}$ with finite risk,

$$\mathcal{L}_{\mathrm{FM}}(\boldsymbol{v}) = \mathcal{L}_{\mathrm{FM}}(\boldsymbol{v}_P^\star) + \mathbb{E}_{t, \boldsymbol{X}_t} \left[ \|\boldsymbol{v}(\boldsymbol{X}_t, t) - \boldsymbol{v}_P^\star(\boldsymbol{X}_t, t)\|^2 \right]. \tag{6}$$

Thus, an optimally trained deterministic field predicts the conditional mean of the velocities associated with all interpolation pairs that can pass through the same state. In this paper, $\boldsymbol{v}_\theta$ is pre-trained with Eq. 4, frozen, and treated as an approximation to $\boldsymbol{v}_P^\star$.

## 3.2 Likelihood Evaluation with Flow Priors

Let $p_1^{\boldsymbol{v}_\theta}(\boldsymbol{x}_1)$ denote the probability density at $t = 1$ induced by the regular ODE flow in Eq. 2. Using the instantaneous change-of-variables formula (Chen et al., 2018; Grathwohl et al., 2019), its log-likelihood can be computed as

$$\log p_1^{\boldsymbol{v}_\theta}(\boldsymbol{x}_1) = \log p_{\text{init}}(\boldsymbol{x}_0) - \int_0^1 \text{Tr}(\nabla_{\boldsymbol{x}} \boldsymbol{v}_\theta(\boldsymbol{x}_s, s))\, \mathrm{d}s, \tag{7}$$

where $\boldsymbol{x}_s$ is the ODE trajectory ending at $\boldsymbol{x}_1$ and $\boldsymbol{x}_0$ is obtained by solving the flow backward to $t = 0$. This formula applies when the flow is sufficiently regular and the induced endpoint law admits a density.

Directly optimizing Eq. 7 for every learnable latent is computationally expensive because it requires an ODE solve and Jacobian-trace evaluation. Our method instead reuses the conditional FM regression objective with the learnable latent treated as a candidate endpoint. Section 4.3 shows that the resulting population objective is an exact multiscale Fisher prior energy and that, for isotropic Gaussian priors, it recovers the NLL up to an affine transformation. This provides a tractable likelihood-oriented alignment signal without evaluating Eq. 7 during downstream optimization.

## 4 Method

In this paper, we aim to align a learnable latent space, whose latents are denoted by $\boldsymbol{z}$, to a prior distribution $p_{\text{prior}}$. We first describe the overall pipeline in Sec. 4.1. Our method leverages a pre-trained FM model to capture the prior dynamics and subsequently align the latents $\boldsymbol{z}$ with them. We then provide an intuitive explanation in Sec. 4.2 and a formal score-space analysis in Sec. 4.3.

### 4.1 Pipeline

In an autoencoder (AE) with encoder $E_\phi$ and decoder $D_\psi$, we want the latent codes of the encoder to conform to the structure of features from a pre-trained feature extractor. More generally, let $\boldsymbol{z} \in \mathbb{R}^{d_1}$ denote a sample from a learnable latent space produced by a parametric model $E_\phi$, and let $\boldsymbol{x} \in \mathbb{R}^{d_2}$ be a sample from a prior feature distribution $p_{\text{prior}}$. Our objective is to train $E_\phi$ such that its output distribution $p_\phi(\boldsymbol{z})$ is shaped by $p_{\text{prior}}$. At the pointwise level, this goal favors latent codes assigned low prior energy and, for continuous priors, high prior likelihood; at the distribution level, it encourages agreement between the multiscale geometry of the latent and prior distributions.

**Addressing the Dimension Mismatch** A challenge arises if the latent-space dimension $d_1$ differs from the prior-feature dimension $d_2$. To address this, we employ fixed, non-learnable linear projections to map prior features $\boldsymbol{x}$ from $\mathbb{R}^{d_2}$ to $\mathbb{R}^{d_1}$. For simplicity, we continue to denote the projected features and their distribution by $\boldsymbol{x}$ and $p_{\text{prior}}$, respectively. We consider three alternative projection operators: *Random Projection*, *Average Pooling*, and *PCA*. We ablate these methods in Sec. 5.3 and select random projection as the default because of its simplicity and empirical effectiveness.

The random projection is defined by a matrix $\boldsymbol{W} \in \mathbb{R}^{d_1 \times d_2}$ whose entries are sampled from $\mathcal{N}(0, 1/d_2)$. For any fixed $\boldsymbol{a} \in \mathbb{R}^{d_2}$,

$$\mathbb{E}\left[\|\boldsymbol{W}\boldsymbol{a}\|^2\right] = \frac{d_1}{d_2}\|\boldsymbol{a}\|^2. \tag{8}$$

Thus, the projection preserves Euclidean geometry in expectation up to the deterministic global factor $d_1/d_2$. Equivalently, $\widetilde{\boldsymbol{W}} = \sqrt{d_2/d_1}\, \boldsymbol{W}$ has the standard unit-scale Johnson–Lindenstrauss normalization (Johnson et al., 1984). We retain the implemented scaling because, for normalized features with approximately uncorrelated unit-variance components, the projected components also have approximately unit variance.

**Flow Prior Estimation** With the projected prior features $\boldsymbol{x} \sim p_{\text{prior}}$, we first train an FM model $\boldsymbol{v}_\theta : \mathbb{R}^{d_1} \times [0,1] \to \mathbb{R}^{d_1}$ using Eq. 4, where $\boldsymbol{x}_0 \sim \mathcal{N}(\boldsymbol{0}, \boldsymbol{I})$ and $\boldsymbol{x}_1$ is replaced by an independent sample $\boldsymbol{x}$ from $p_{\text{prior}}$. After training, the parameters $\theta$ are frozen. The learned field approximates the population conditional velocity in Eq. 5 and captures the prior distribution through its Gaussian-to-prior interpolation dynamics. For continuous endpoint distributions, the same field can also define an approximate generative ODE. The alignment method itself only requires evaluations of the frozen field and therefore applies equally to continuous feature distributions and sample-defined atomic endpoints such as VQ codebook embeddings.

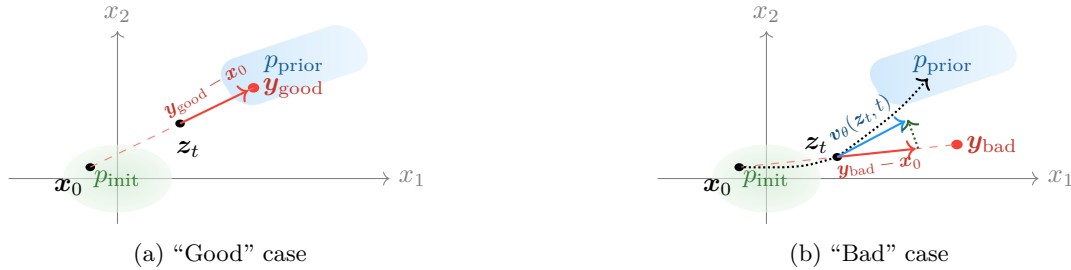

(a) "Good" case            (b) "Bad" case

Figure 2: Intuitive illustration of latent-space alignment via the frozen flow prior, best viewed in color. (a) A prior-compatible endpoint $\boldsymbol{y}_{\text{good}}$ yields a small mismatch between its straight-path velocity and the conditional velocity predicted by the pre-trained flow model. (b) An endpoint $\boldsymbol{y}_{\text{bad}}$ outside the high-density prior regions yields a larger mismatch. Minimizing the alignment loss steers $\boldsymbol{y}_{\text{bad}}$ toward prior-compatible low-energy regions (blue dotted arrow).

**Latent Space Regularization** Once $\boldsymbol{v}_\theta$ is trained and frozen, we use it to regularize the learnable latents $\boldsymbol{z}$. For each $\boldsymbol{z}$ produced by $E_\phi$, we incorporate the following alignment objective:

$$\mathcal{L}_{\text{align}}(\boldsymbol{z}; \theta) = \mathbb{E}_{t \sim \mathcal{U}[0,1], \, \boldsymbol{x}_0 \sim p_{\text{init}}} \left[ \| \boldsymbol{v}_\theta((1-t)\boldsymbol{x}_0 + t\boldsymbol{z}, t) - (\boldsymbol{z} - \boldsymbol{x}_0) \|^2 \right]. \tag{9}$$

Here, $p_{\text{init}} = \mathcal{N}(\boldsymbol{0}, \boldsymbol{I})$ is the same $d_1$-dimensional base distribution used to train $\boldsymbol{v}_\theta$. The objective has the same regression form as standard FM, but the endpoint is now a learnable latent rather than a fixed prior sample. Section 4.3 shows that, under the population-optimal FM field, Eq. 9 is an exact multiscale Fisher prior energy. In particular, for isotropic Gaussian priors it is an affine transformation of the exact NLL. This establishes Eq. 9 as a theoretically grounded and computationally tractable likelihood surrogate.

The key insight is that the pre-trained velocity field encapsulates the conditional dynamics associated with paths terminating in the prior distribution. By minimizing $\mathcal{L}_{\text{align}}$, we penalize latent endpoints whose noisy interpolation paths are inconsistent with these learned dynamics. The resulting energy guides latents toward prior-compatible, high-likelihood regions without requiring explicit density evaluation, Jacobian traces, or ODE solving.

### 4.2 Intuitive Explanation

Our alignment method leverages the pre-trained flow model $\boldsymbol{v}_\theta$ as an expert on the prior feature distribution $p_{\text{prior}}$. A well-trained FM model predicts the conditional mean velocity associated with Gaussian-to-prior interpolation paths. Equivalently, it induces an endpoint predictor

$$\widehat{\boldsymbol{x}}_\theta(\boldsymbol{y}, t) = \boldsymbol{y} + (1-t)\boldsymbol{v}_\theta(\boldsymbol{y}, t). \tag{10}$$

For a candidate endpoint $\boldsymbol{z}$ and a base sample $\boldsymbol{x}_0$, define

$$\boldsymbol{y}_t = (1-t)\boldsymbol{x}_0 + t\boldsymbol{z}. \tag{11}$$

The velocity residual in Eq. 9 satisfies the exact identity

$$\boldsymbol{v}_\theta(\boldsymbol{y}_t, t) - (\boldsymbol{z} - \boldsymbol{x}_0) = \frac{\widehat{\boldsymbol{x}}_\theta(\boldsymbol{y}_t, t) - \boldsymbol{z}}{1-t}, \qquad t < 1. \tag{12}$$

Therefore,

$$\mathcal{L}_{\text{align}}(\boldsymbol{z}; \theta) = \mathbb{E}_{t, \boldsymbol{x}_0} \left[ \frac{\|\widehat{\boldsymbol{x}}_\theta((1-t)\boldsymbol{x}_0 + t\boldsymbol{z}, t) - \boldsymbol{z}\|^2}{(1-t)^2} \right]. \tag{13}$$

Equation 13 shows that the alignment loss perturbs a candidate latent across multiple noise levels and asks the prior-trained endpoint predictor to recover that candidate. A prior-compatible latent typically forms a stable endpoint for this denoising rule and receives low energy, whereas an incompatible latent produces a larger endpoint residual. Minimizing the loss therefore moves learnable latents toward regions endorsed by the prior dynamics, as illustrated in Fig. 2.

### 4.3 Theoretical Foundation: A Multiscale Likelihood Surrogate

Let $P = p_{\text{prior}}$ be a probability measure on $\mathbb{R}^{d_1}$ with finite second moment. Let $\boldsymbol{X} \sim P$ and $\boldsymbol{\epsilon} \sim \mathcal{N}(\boldsymbol{0}, \boldsymbol{I})$ be independent, define $\sigma_t = 1 - t$, and set

$$\boldsymbol{Y}_t = \sigma_t \boldsymbol{\epsilon} + t\boldsymbol{X}. \tag{14}$$

Let $P_t$ denote the law of $\boldsymbol{Y}_t$ and $p_t$ its smooth positive density for $t \in (0, 1)$. All per-time identities below hold for $t \in (0, 1)$; integrated identities hold whenever the displayed integrals are finite.

For smooth positive densities $r$ and $s$, define the relative Fisher divergence

$$\mathcal{I}(R\|S) = \mathbb{E}_{\boldsymbol{Y} \sim R} \left[ \|\nabla \log r(\boldsymbol{Y}) - \nabla \log s(\boldsymbol{Y})\|^2 \right]. \tag{15}$$

**Proposition 1** (Population score and endpoint representation). *The population-optimal conditional FM field satisfies*

$$\boldsymbol{v}_P^\star(\boldsymbol{y}, t) = \frac{\boldsymbol{y}}{t} + \frac{\sigma_t}{t} \nabla_{\boldsymbol{y}} \log p_t(\boldsymbol{y}). \tag{16}$$

*Its endpoint predictor is the Bayes conditional mean:*

$$\boldsymbol{y} + \sigma_t \boldsymbol{v}_P^\star(\boldsymbol{y}, t) = \mathbb{E}[\boldsymbol{X} \mid \boldsymbol{Y}_t = \boldsymbol{y}]. \tag{17}$$

The result follows from the Gaussian score identity and is proved in Appendix A.

For a fixed candidate $\boldsymbol{z}$, define the Gaussian probe

$$K_t^{\boldsymbol{z}} = \mathcal{N}(t\boldsymbol{z}, \sigma_t^2 \boldsymbol{I}), \tag{18}$$

with density $k_t^{\boldsymbol{z}}$, and define the ideal per-time alignment energy

$$\ell_t^\star(\boldsymbol{z}) = \mathbb{E}_{\boldsymbol{\epsilon}} \left[ \|\boldsymbol{v}_P^\star(\sigma_t \boldsymbol{\epsilon} + t\boldsymbol{z}, t) - (\boldsymbol{z} - \boldsymbol{\epsilon})\|^2 \right]. \tag{19}$$

**Proposition 2** (Pointwise Fisher characterization). *For every fixed $\boldsymbol{z} \in \mathbb{R}^{d_1}$ and $t \in (0, 1)$,*

$$\ell_t^\star(\boldsymbol{z}) = \frac{\sigma_t^2}{t^2} \mathcal{I}(K_t^{\boldsymbol{z}} \| P_t). \tag{20}$$

*Consequently, the ideal alignment energy is*

$$\mathcal{L}_{\text{align}}^\star(\boldsymbol{z}) = \int_0^1 \frac{(1-t)^2}{t^2} \mathcal{I}(K_t^{\boldsymbol{z}} \| P_t) \, dt. \tag{21}$$

Proposition 2 gives an exact interpretation of the alignment loss as a multiscale prior energy. At every interpolation time, it compares the score of the Gaussian probe centered at the candidate endpoint with the score of the Gaussian-smoothed prior. Low energy therefore identifies candidates whose local noisy neighborhoods agree with the prior geometry across scales.

The relationship admits a closed-form characterization for Gaussian priors and becomes exact, up to an affine transformation, for isotropic Gaussian priors.

**Proposition 3** (Gaussian prior geometry). *Let $P = \mathcal{N}(\boldsymbol{\mu}, \boldsymbol{\Sigma})$ with $\boldsymbol{\Sigma} \succ \mathbf{0}$. Then there exists a constant $C_P$, independent of $\boldsymbol{z}$, such that*

$$\mathcal{L}^{\star}_{\text{align}}(\boldsymbol{z}) = C_P + \frac{\pi}{4}(\boldsymbol{z} - \boldsymbol{\mu})^{\top} \boldsymbol{\Sigma}^{-1/2} (\boldsymbol{z} - \boldsymbol{\mu}). \tag{22}$$

*In particular, if $\boldsymbol{\Sigma} = s^2 \boldsymbol{I}$, then for constants $a_s = \pi s / 2 > 0$ and $b_s$,*

$$\mathcal{L}^{\star}_{\text{align}}(\boldsymbol{z}) = a_s \big[ -\log p_{\text{prior}}(\boldsymbol{z}) \big] + b_s. \tag{23}$$

Thus, for every isotropic Gaussian prior, minimizing the ideal alignment loss is exactly equivalent to minimizing the NLL. For a general Gaussian prior, the energy has the same center and principal directions as the Gaussian NLL, with a softened Mahalanobis weighting $\boldsymbol{\Sigma}^{-1/2}$. The controlled experiments in Sec. 5.1 show that the same likelihood-surrogate behavior extends empirically to multimodal and non-Gaussian priors.

We next characterize the objective averaged over an aggregate latent distribution. Let $\boldsymbol{Z} \sim Q$, where $Q$ has finite second moment, be independent of $\boldsymbol{\epsilon}$ and define

$$\boldsymbol{Y}^{Q}_{t} = \sigma_t \boldsymbol{\epsilon} + t\boldsymbol{Z}, \qquad Q_t = \text{Law}(\boldsymbol{Y}^{Q}_{t}). \tag{24}$$

Define the Gaussian-channel minimum mean-squared error

$$\text{mmse}_Q(t) = \mathbb{E}\left[ \| \boldsymbol{Z} - \mathbb{E}[\boldsymbol{Z} \mid \boldsymbol{Y}^{Q}_{t}] \|^2 \right]. \tag{25}$$

**Proposition 4** (Aggregate Fisher–MMSE decomposition). *For every $t \in (0, 1)$,*

$$\mathbb{E}_{\boldsymbol{Z} \sim Q}[\ell^{\star}_t(\boldsymbol{Z})] = \frac{\sigma_t^2}{t^2} \mathcal{I}(Q_t \| P_t) + \frac{\text{mmse}_Q(t)}{\sigma_t^2}. \tag{26}$$

*Hence,*

$$\mathbb{E}_{\boldsymbol{Z} \sim Q}[\mathcal{L}^{\star}_{\text{align}}(\boldsymbol{Z})] = \int_0^1 \left[ \frac{(1-t)^2}{t^2} \mathcal{I}(Q_t \| P_t) + \frac{\text{mmse}_Q(t)}{(1-t)^2} \right] \mathrm{d}t. \tag{27}$$

The first term in Eq. 27 is a proper multiscale distributional discrepancy: it is nonnegative and vanishes only when $Q = P$ under the regularity conditions in Appendix A. The second term promotes endpoint recoverability from noisy interpolants and therefore favors a compact, readily modelable latent representation. This decomposition explains the two effects observed in our experiments: the loss transfers prior structure through the Fisher term while simplifying the latent space through the recoverability term. Because both terms are optimized jointly, Eq. 9 is an alignment regularizer rather than a standalone divergence whose unique minimizer is $Q = P$.

Finally, the practical loss uses the learned field $\boldsymbol{v}_\theta$ instead of $\boldsymbol{v}^{\star}_P$. Define

$$J_\theta(Q) = \mathbb{E}_{\boldsymbol{Z} \sim Q}[\mathcal{L}_{\text{align}}(\boldsymbol{Z}; \theta)], \tag{28}$$

$$J^{\star}(Q) = \mathbb{E}_{\boldsymbol{Z} \sim Q}[\mathcal{L}^{\star}_{\text{align}}(\boldsymbol{Z})], \tag{29}$$

$$\Delta_\theta(Q) = \mathbb{E}_{t, \boldsymbol{Z}, \boldsymbol{\epsilon}} \left[ \| \boldsymbol{v}_\theta(\sigma_t \boldsymbol{\epsilon} + t\boldsymbol{Z}, t) - \boldsymbol{v}^{\star}_P(\sigma_t \boldsymbol{\epsilon} + t\boldsymbol{Z}, t) \|^2 \right]. \tag{30}$$

**Proposition 5** (Approximation stability). *Whenever the three quantities above are finite,*

$$\left| \sqrt{J_\theta(Q)} - \sqrt{J^{\star}(Q)} \right| \leq \sqrt{\Delta_\theta(Q)}. \tag{31}$$

Equation 31 shows that the learned objective inherits the ideal multiscale prior-energy interpretation up to the FM regression error on the interpolants visited by the current latents. Together, Propositions 2–5 provide the theoretical basis for using a frozen flow prior as an efficient likelihood-oriented latent-space alignment objective.

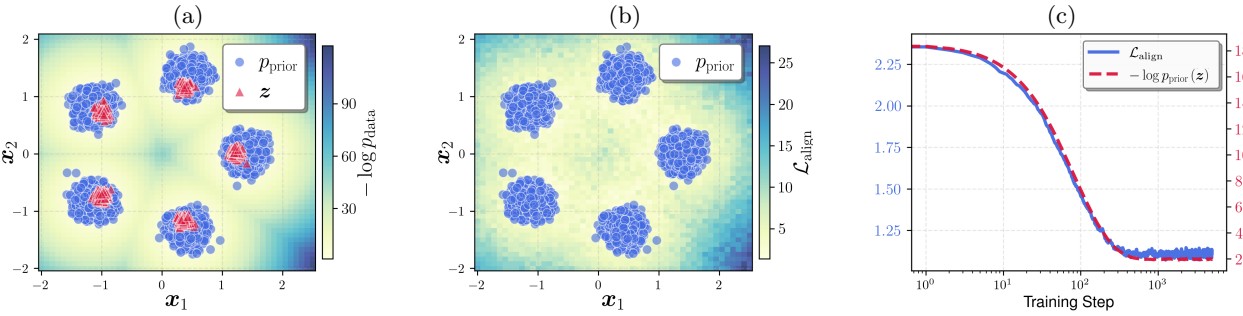

Figure 3: Illustration with a mixture-of-Gaussians prior. (a) Aligned latent variables $\boldsymbol{z}$ (red triangles) concentrate in low-NLL regions of $p_{\mathrm{prior}}$ (blue dots; the heatmap shows $-\log p_{\mathrm{prior}}$). (b) The alignment-loss heatmap closely mirrors the NLL landscape, with prior samples concentrated in low-$\mathcal{L}_{\mathrm{align}}$ regions. (c) $\mathcal{L}_{\mathrm{align}}$ (blue solid) and the exact NLL $-\log p_{\mathrm{prior}}(\boldsymbol{z})$ (red dashed) decline together during optimization, demonstrating that the alignment loss is an effective NLL surrogate in this controlled setting.

## 5 Experiments

This section presents an empirical validation of the proposed alignment method with flow priors. The investigation starts with an illustrative experiment in Sec. 5.1. Subsequently, large-scale experiments are conducted on image generation tasks using the ImageNet dataset, as detailed in Sec. 5.2. In Sec. 5.3, we conduct ablation studies of the proposed method.

### 5.1 Toy Examples

We present a toy example as an illustrative experiment in a 2D setting. The prior distribution, denoted $p_{\mathrm{prior}}$, is configured as a mixture of five isotropic Gaussians. Following the method outlined in Sec. 4.1, we first train an FM model $\boldsymbol{v}_\theta$ on interpolation paths between a standard Normal $\mathcal{N}(\boldsymbol{0}, \boldsymbol{I})$ and samples from $p_{\mathrm{prior}}$. This FM model is implemented by a multi-layer perceptron (MLP) incorporating adaptive layer normalization for time modulation (Peebles & Xie, 2023). Upon completion of training, the parameters $\theta$ are frozen. Subsequently, instead of a parameterized model $E_\phi$, we directly initialize a set of learnable 2D variables $\boldsymbol{z}$ and optimize them by minimizing the alignment loss $\mathcal{L}_{\mathrm{align}}(\boldsymbol{z}; \theta)$.

The results are presented in Fig. 3. Figure 3(a) compares samples from the prior distribution $p_{\mathrm{prior}}$ (blue points) with the optimized variables $\boldsymbol{z}$ (red triangles). The background visualizes the analytically computed negative log-likelihood (NLL) of $p_{\mathrm{prior}}$. The optimized variables successfully converge to high-likelihood regions of the prior. Figure 3(b) displays the landscape of the alignment loss $\mathcal{L}_{\mathrm{align}}$, estimated numerically with $\boldsymbol{v}_\theta$. The alignment-loss landscape closely mirrors the NLL surface in panel (a), and samples from $p_{\mathrm{prior}}$ are concentrated in low-$\mathcal{L}_{\mathrm{align}}$ regions. This confirms that the learned flow prior converts the target density structure into a tractable pointwise energy. Figure 3(c) illustrates $\mathcal{L}_{\mathrm{align}}$ (blue solid line) and the exact NLL $-\log p_{\mathrm{prior}}(\boldsymbol{z})$ (red dashed line) during optimization. The two quantities exhibit a strong positive correlation and decrease concomitantly throughout training, demonstrating that $\mathcal{L}_{\mathrm{align}}$ serves as an effective NLL surrogate in this controlled setting.

The same pattern holds across a broader set of two-dimensional priors, including a grid of Gaussians, two moons, concentric rings, a spiral, and a Swiss roll. For the grid prior, the NLL is computed analytically; for the nonparametric priors, we estimate the NLL with Gaussian-kernel density estimation using 100,000 samples and bandwidth 0.1. Across these cases, minimizing $\mathcal{L}_{\mathrm{align}}$ moves the optimized variables from random initialization toward high-density regions of the target distribution, and the alignment-loss landscape remains closely aligned with the analytic or KDE-estimated NLL. Additional landscapes and optimization trajectories are provided in Appendix B.

## 5.2 Image Generation

Prior work has demonstrated that aligning the latent space of AEs with semantic encoders can enhance generative model performance (Chen et al., 2025b; Yao et al., 2025; Chen et al., 2024; Qiu et al., 2025). To validate this observation and further showcase the efficacy of our proposed method, we conduct large-scale image generation experiments on the ImageNet-1K (Deng et al., 2009) dataset at $256 \times 256$ resolution.

**Implementation Details**   Our AE architecture employs two Vision Transformer (ViT)-Large (Dosovitskiy et al., 2021) models, each with 391M parameters, serving as the latent encoder and decoder, respectively. The encoder maps input images to a latent space of 64 tokens, each with dimension 32, striking a balance between reconstruction quality and computational efficiency. We impose *token-level* alignment on the latents. Accordingly, the distributional analysis in Sec. 4.3 applies to the marginal distribution of latent tokens pooled over images and token positions. The alignment loss on the latents is set to $\lambda = 0.01$ by default. We also incorporate conventional reconstruction loss, perceptual loss, and adversarial loss on the pixel outputs (Esser et al., 2021; Rombach et al., 2022).

For the prior distribution $p_{\text{prior}}$, we investigate four distinct variants: *low-level* visual features from a VAE (Rombach et al., 2022), continuous *semantic* visual features from DinoV2 (Oquab et al., 2024), *discrete* visual codebook embeddings from LlamaGen VQ (Sun et al., 2024; Van Den Oord et al., 2017), and *textual* embeddings from Qwen (Bai et al., 2023). Their feature dimensions are 32, 768, 8, and 896, respectively. The flow prior for each feature distribution is a 6-layer MLP with 1024 hidden units, GELU activations, and AdaLN time modulation, trained for 1 million steps using AdamW (Loshchilov & Hutter, 2019) with the conditional FM objective in Eq. 4. The autoencoder follows a ViT-Large encoder–decoder design with a DINOv2-initialized encoder and a randomly initialized decoder. We use a DINOv2-based discriminator, VGG perceptual loss with a warmup period, EMA, BF16 mixed precision, and gradient clipping. The autoencoder is trained for $50k$ steps; the MAR-B generator is trained for $250k$ steps with qk-norm and a flow head. All experiments use the same hardware and identical training settings except for the latent prior, so the comparisons isolate the effect of alignment. Detailed hyperparameters are provided in Appendix C.

The additional cost of the flow prior is small relative to the rest of the pipeline. On a single node with 8 GPUs, training the MLP flow prior for 1M steps takes a few hours, whereas training the autoencoder and MAR-B generator takes roughly 50 and 70 hours, respectively. During autoencoder training, the alignment term requires only one frozen MLP forward pass per iteration and does not noticeably affect throughput compared with a KL-regularized baseline.

**Alignment Results**   Analogous to the toy example, we compare the alignment loss $\mathcal{L}_{\text{align}}$ with an NLL proxy obtained from $k$-nearest neighbors, using $k = 5$. For a continuous distribution in a fixed dimension $d$, the standard $k$-NN density estimate satisfies $\widehat{p}(\boldsymbol{z}) \propto r_k(\boldsymbol{z})^{-d}$ up to factors independent of $\boldsymbol{z}$. Consequently,

$$-\log \widehat{p}(\boldsymbol{z}) = d \log r_k(\boldsymbol{z}) + \text{constant}, \tag{32}$$

where $r_k(\boldsymbol{z})$ is the Euclidean distance to the $k$th nearest prior sample. Since $d$ is fixed within each run, $\log r_k(\boldsymbol{z})$ is proportional to the corresponding $k$-NN NLL proxy up to a positive scale and an additive constant. We index the prior samples using Faiss (Douze et al., 2024); during training, we periodically sample 10k latent tokens and average $\log r_k(\boldsymbol{z})$.

The results are presented in Fig. 4. A strong correlation is observed between the alignment loss $\mathcal{L}_{\text{align}}$ and the $k$-NN NLL proxy $\log r_k(\boldsymbol{z})$ for the continuous-prior variants. This finding corroborates the conclusion from the toy experiments that $\mathcal{L}_{\text{align}}$ serves as an effective surrogate for NLL trends under implicitly defined priors. The VQ prior is an atomic codebook-embedding distribution, so $\log r_k(\boldsymbol{z})$ is interpreted as codebook-neighborhood proximity rather than a continuous-density NLL; despite the GAN collapse in this low-dimensional variant, the alignment and proximity curves retain the same overall trend before instability. Crucially, the proposed method captures prior structure across different forms and modalities, including continuous visual features, discrete codebook embeddings, and textual representations, even on ImageNet with a high-capacity ViT-Large autoencoder.

To verify that this behavior comes from the alignment objective, we also train an autoencoder with the same Dino prior target but remove the alignment loss. In that setting, the $k$-NN distance to the prior does not improve and can drift away during training, confirming that the observed alignment is induced by the proposed objective. The corresponding diagnostic plot is provided in Appendix C.

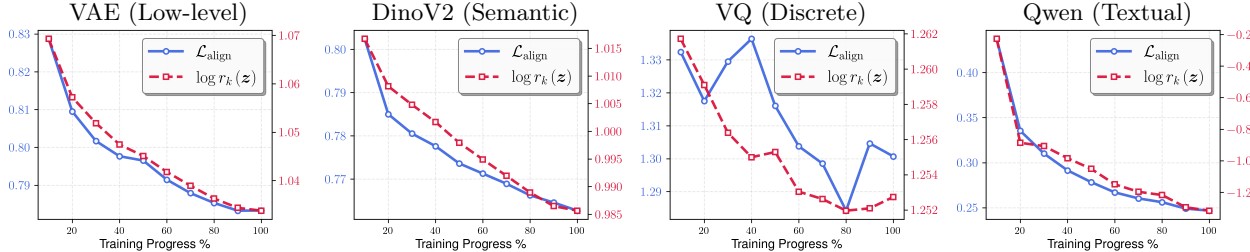

Figure 4: Aligning autoencoders on ImageNet-1K with different prior distributions. The alignment loss $\mathcal{L}_{\text{align}}$ (blue solid) and the $k$-NN statistic $\log r_k(\boldsymbol{z})$ (red dashed) show strong agreement throughout training. For the continuous-prior variants, $\log r_k(\boldsymbol{z})$ is proportional, up to a positive scale and additive constant, to the corresponding fixed-dimensional $k$-NN NLL proxy. For the atomic VQ prior, it measures codebook-neighborhood proximity.

**Generation Results** After demonstrating effective latent-space alignment, we investigate its impact on generative model performance. We evaluate both reconstruction and generation capabilities on ImageNet using the MAR-B (Li et al., 2024a) architecture. For MAR-B, we incorporate qk-norm (Dehghani et al., 2023) and replace the diffusion head with a flow head to ensure stable training. We choose flow-based MAR-B because its objective does not specifically favor a continuous Gaussian-like latent structure in the same way as diffusion models (Song et al., 2021; Dhariwal & Nichol, 2021; Karras et al., 2022; Nichol & Dhariwal, 2021; Rombach et al., 2022). To ensure an "apples-to-apples" comparison, configurations and hardware remain identical across all experiments, with the only difference being the specific AE used for each alignment variant.

Table 1: ImageNet $256 \times 256$ conditional generation using MAR-B. All models are trained and evaluated using identical settings. The CFG scale is tuned for KL and kept the same for others.

| Autoencoder | rFID↓ | PSNR↑ | w/o CFG | | | | w/ CFG | | | |
|---|---|---|---|---|---|---|---|---|---|---|
| | | | FID↓ | IS↑ | Pre.↑ | Rec.↑ | FID↓ | IS↑ | Pre.↑ | Rec.↑ |
| AE | 1.13 | 20.20 | 15.08 | 86.37 | **0.60** | **0.59** | 5.26 | 237.60 | 0.56 | 0.65 |
| KL | 1.65 | 22.59 | 12.94 | 91.86 | 0.60 | 0.58 | 5.29 | 200.85 | 0.57 | 0.65 |
| SoftVQ | **0.61** | 23.00 | 13.30 | 93.40 | 0.60 | 0.57 | 6.09 | 198.53 | **0.58** | 0.61 |
| Low-level (VAE) | 1.22 | 22.31 | 12.04 | 98.66 | 0.56 | 0.57 | 5.02 | 240.03 | 0.56 | 0.62 |
| Semantic (Dino) | 1.26 | 23.07 | **11.47** | 101.74 | 0.59 | 0.59 | **4.87** | **250.38** | 0.54 | 0.67 |
| Discrete (VQ) | 2.99 | 22.32 | 24.63 | 48.17 | 0.55 | 0.53 | 10.04 | 119.64 | 0.47 | 0.65 |
| Textual (Qwen) | 0.85 | **23.12** | 11.89 | **102.23** | 0.55 | 0.57 | 6.56 | 262.89 | 0.49 | **0.69** |

The results are presented in Tab. 1. Reconstruction performance is measured by rFID (Heusel et al., 2017) and PSNR on the ImageNet validation set. Generation performance is assessed using FID, IS (Salimans et al., 2016), Precision, and Recall on 50k generated samples and the validation set, both with and without classifier-free guidance (CFG) (Ho & Salimans, 2022). AE does not impose any prior, KL uses a standard Gaussian prior, and SoftVQ performs per-sample alignment with Dino features; all other priors match those used in the alignment experiments. Our key findings are:

*1) Alignment vs. Reconstruction Trade-off:* Latent-space alignment typically degrades reconstruction quality (rFID and PSNR) compared with vanilla AEs because the additional structure constrains representational capacity. SoftVQ excels among aligned methods because of its sample-level alignment. *2) Alignment Enhances Generation:* All stable continuous flow-prior variants improve unguided FID and IS over the vanilla AE, confirming that a structured latent space is easier for the downstream generator to model. The semantic Dino prior also achieves the best guided FID. Prior complexity alone is not decisive: textual Qwen embeddings

Table 2: Ablation studies on ImageNet $256 \times 256$ for different configurations using autoencoders regularized by textual features (Qwen). We use a shorter training schedule when ablating weight.

<table>
<tr><td colspan="5">(a) Downsampling Methods</td><td colspan="5">(b) Alignment Loss Weight</td></tr>
<tr><td>Method</td><td>rFID↓</td><td>PSNR↑</td><td>FID↓</td><td>IS↑</td><td>Weight $\lambda$</td><td>rFID↓</td><td>PSNR↑</td><td>FID↓</td><td>IS↑</td></tr>
<tr><td>Random Proj.</td><td>**0.85**</td><td>23.12</td><td>**11.89**</td><td>**102.23**</td><td>0.001</td><td>**0.89**</td><td>22.78</td><td>17.57</td><td>75.20</td></tr>
<tr><td>Avg. Pooling</td><td>0.94</td><td>22.98</td><td>16.06</td><td>60.37</td><td>0.005</td><td>1.02</td><td>22.98</td><td>16.93</td><td>78.01</td></tr>
<tr><td>PCA</td><td>0.87</td><td>**23.14**</td><td>14.95</td><td>83.59</td><td>0.01</td><td>1.31</td><td>**23.12**</td><td>13.67</td><td>82.13</td></tr>
<tr><td></td><td></td><td></td><td></td><td></td><td>0.05</td><td>1.81</td><td>21.82</td><td>**12.30**</td><td>**92.48**</td></tr>
</table>

remain competitive with richer visual features. *3) Optimal Prior Selection Is Open:* Low-dimensional discrete VQ features underperform, while cross-modal Qwen alignment demonstrates transferable structural benefits. Taken together, these observations suggest that prior dimensionality, smoothness, and semantic organization jointly matter. Strong alignment can simplify the latent distribution and improve modelability, while excessive simplification can reduce the diversity and fine detail required for high-quality generation.

## 5.3 Ablation Study

**Downsampling Operators** We ablate the downsampling operators in Tab. 2(a). We adopt the same settings as in Tab. 1 using textual embeddings from Qwen as the prior distribution. Despite all being linear downsampling operators, PCA and average pooling perform worse than random projection. Equation 8 shows that the implemented random projection preserves squared Euclidean geometry in expectation up to a fixed global scale. The ablation further demonstrates that this simple projection retains the distributional structure needed for effective alignment better than the two alternatives in our setting.

**Alignment Loss Weight** We apply different strengths of regularization by altering the alignment-loss weight $\lambda$ in Tab. 2(b). Larger weights impose stronger prior energy, leading to worse reconstruction but consistently better unguided FID and IS over the reported range. This trend is consistent with the aggregate decomposition in Eq. 27: increasing the alignment weight places greater optimization pressure on the combined Fisher-discrepancy and endpoint-recoverability terms. At the same time, excessive simplification can destabilize adversarial training, as observed for the low-dimensional VQ prior. The alignment weight therefore controls the trade-off between reconstruction capacity and generative modelability.

## 6 Conclusion

This paper introduced a novel method for aligning learnable latent spaces with arbitrary sample-defined prior distributions by leveraging pre-trained flow-matching models as expressive priors. Our approach uses a computationally tractable alignment loss, adapted from the flow-matching objective, to guide latent variables toward prior-compatible, high-likelihood regions while requiring only frozen velocity-field evaluations during downstream training. We established an exact score-space characterization of the objective: under the population-optimal conditional FM field, the pointwise loss is a multiscale relative Fisher energy between candidate-induced Gaussian probes and Gaussian-smoothed prior marginals. For isotropic Gaussian priors, this energy is an affine transformation of the exact NLL; at the aggregate level, its expectation decomposes into a proper multiscale Fisher discrepancy and an endpoint-recoverability term. Controlled toy experiments further demonstrate that the alignment-loss landscape closely approximates analytic or KDE-estimated NLL landscapes across diverse multimodal priors. Large-scale ImageNet experiments show that the framework can transfer structure from visual, discrete, and textual priors and improve downstream generation for stable continuous-prior variants. Ultimately, this work provides a flexible and powerful framework for incorporating rich distributional priors, paving the way for more structured and interpretable representation learning. A limitation, and also a promising future direction, is that the selection of the optimal prior remains task dependent. While semantic priors are effective for image generation, no single "silver bullet" prior is expected to be optimal across all tasks.

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

# A  Proofs for the Score-Space Characterization

## A.1  Population Flow-Matching Regression

Let $\boldsymbol{X}_0 \sim p_{\text{init}}$ and $\boldsymbol{X}_1 \sim P$ be independent, let $\boldsymbol{X}_t = (1-t)\boldsymbol{X}_0 + t\boldsymbol{X}_1$, and set $\boldsymbol{U} = \boldsymbol{X}_1 - \boldsymbol{X}_0$. For fixed $t$, the squared-loss regression problem is

$$\inf_{\boldsymbol{v}} \mathbb{E}\left[\|\boldsymbol{v}(\boldsymbol{X}_t, t) - \boldsymbol{U}\|^2\right]. \tag{33}$$

The conditional expectation

$$\boldsymbol{v}_P^\star(\boldsymbol{x}, t) = \mathbb{E}[\boldsymbol{U} \mid \boldsymbol{X}_t = \boldsymbol{x}] \tag{34}$$

is its population minimizer. For any measurable $\boldsymbol{v}$ with finite risk, conditioning on $\boldsymbol{X}_t$ gives

$$\mathbb{E}\left[\|\boldsymbol{v}(\boldsymbol{X}_t, t) - \boldsymbol{U}\|^2\right] = \mathbb{E}\left[\|\boldsymbol{v}_P^\star(\boldsymbol{X}_t, t) - \boldsymbol{U}\|^2\right] \tag{35}$$

$$+ \mathbb{E}\left[\|\boldsymbol{v}(\boldsymbol{X}_t, t) - \boldsymbol{v}_P^\star(\boldsymbol{X}_t, t)\|^2\right], \tag{36}$$

because the cross term has zero conditional expectation. Averaging over $t$ proves Eq. 6.

### A.2 Proof of Proposition 1

Let $\sigma_t = 1 - t$, $\boldsymbol{Y}_t = \sigma_t \boldsymbol{\epsilon} + t \boldsymbol{X}$, and let $\varphi_{\sigma_t}$ denote the density of $\mathcal{N}(\boldsymbol{0}, \sigma_t^2 \boldsymbol{I})$. The density of $P_t$ is

$$p_t(\boldsymbol{y}) = \int \varphi_{\sigma_t}(\boldsymbol{y} - t\boldsymbol{x}) \, \mathrm{d}P(\boldsymbol{x}). \tag{37}$$

For $t \in (0, 1)$, differentiation under the integral gives

$$\nabla_{\boldsymbol{y}} \log p_t(\boldsymbol{y}) = -\frac{1}{\sigma_t^2} \left( \boldsymbol{y} - t\mathbb{E}[\boldsymbol{X} \mid \boldsymbol{Y}_t = \boldsymbol{y}] \right) \tag{38}$$

$$= -\frac{1}{\sigma_t} \mathbb{E}[\boldsymbol{\epsilon} \mid \boldsymbol{Y}_t = \boldsymbol{y}]. \tag{39}$$

Consequently,

$$\mathbb{E}[\boldsymbol{X} \mid \boldsymbol{Y}_t = \boldsymbol{y}] = \frac{\boldsymbol{y}}{t} + \frac{\sigma_t^2}{t} \nabla \log p_t(\boldsymbol{y}), \tag{40}$$

and

$$\mathbb{E}[\boldsymbol{\epsilon} \mid \boldsymbol{Y}_t = \boldsymbol{y}] = -\sigma_t \nabla \log p_t(\boldsymbol{y}). \tag{41}$$

Subtracting the two conditional expectations yields

$$\boldsymbol{v}_P^\star(\boldsymbol{y}, t) = \mathbb{E}[\boldsymbol{X} - \boldsymbol{\epsilon} \mid \boldsymbol{Y}_t = \boldsymbol{y}] \tag{42}$$

$$= \frac{\boldsymbol{y}}{t} + \frac{\sigma_t}{t} \nabla \log p_t(\boldsymbol{y}), \tag{43}$$

which proves Eq. 16. Finally,

$$\boldsymbol{y} + \sigma_t \boldsymbol{v}_P^\star(\boldsymbol{y}, t) = \boldsymbol{y} + \sigma_t \mathbb{E}[\boldsymbol{X} - \boldsymbol{\epsilon} \mid \boldsymbol{Y}_t = \boldsymbol{y}] \tag{44}$$

$$= \mathbb{E}[t\boldsymbol{X} + \sigma_t \boldsymbol{\epsilon} + \sigma_t \boldsymbol{X} - \sigma_t \boldsymbol{\epsilon} \mid \boldsymbol{Y}_t = \boldsymbol{y}] \tag{45}$$

$$= \mathbb{E}[\boldsymbol{X} \mid \boldsymbol{Y}_t = \boldsymbol{y}], \tag{46}$$

proving Eq. 17.

### A.3 Proof of Proposition 2

For fixed $\boldsymbol{z}$, the Gaussian probe $K_t^{\boldsymbol{z}} = \mathcal{N}(t\boldsymbol{z}, \sigma_t^2 \boldsymbol{I})$ has score

$$\nabla \log k_t^{\boldsymbol{z}}(\boldsymbol{y}) = -\frac{\boldsymbol{y} - t\boldsymbol{z}}{\sigma_t^2}. \tag{47}$$

Along $\boldsymbol{y} = \sigma_t \boldsymbol{\epsilon} + t\boldsymbol{z}$,

$$\nabla \log k_t^{\boldsymbol{z}}(\boldsymbol{y}) = -\frac{\boldsymbol{\epsilon}}{\sigma_t}. \tag{48}$$

Using Proposition 1,

$$\boldsymbol{v}_P^\star(\boldsymbol{y}, t) - (\boldsymbol{z} - \boldsymbol{\epsilon}) \tag{49}$$

$$= \frac{\boldsymbol{y}}{t} + \frac{\sigma_t}{t} \nabla \log p_t(\boldsymbol{y}) - \boldsymbol{z} + \boldsymbol{\epsilon} \tag{50}$$

$$= -\frac{\sigma_t}{t} \left( \nabla \log k_t^{\boldsymbol{z}}(\boldsymbol{y}) - \nabla \log p_t(\boldsymbol{y}) \right). \tag{51}$$

Taking the squared norm and expectation with respect to $\boldsymbol{y} \sim K_t^{\boldsymbol{z}}$ gives

$$\ell_t^\star(\boldsymbol{z}) = \frac{\sigma_t^2}{t^2} \mathcal{I}(K_t^{\boldsymbol{z}} \| P_t), \tag{52}$$

which is Eq. 20. Integrating over $t$ gives Eq. 21 whenever the integral is finite.

### A.4 Proof of Proposition 3

Let $P = \mathcal{N}(\boldsymbol{\mu}, \boldsymbol{\Sigma})$ with $\boldsymbol{\Sigma} \succ \mathbf{0}$, and define

$$C_t = t^2 \boldsymbol{\Sigma} + \sigma_t^2 \boldsymbol{I}. \tag{53}$$

Then $P_t = \mathcal{N}(t\boldsymbol{\mu}, \boldsymbol{C}_t)$. For $\boldsymbol{y} = t\boldsymbol{z} + \sigma_t \boldsymbol{\epsilon}$,

$$\nabla \log k_t^{\boldsymbol{z}}(\boldsymbol{y}) = -\frac{\boldsymbol{\epsilon}}{\sigma_t}, \tag{54}$$

$$\nabla \log p_t(\boldsymbol{y}) = -\boldsymbol{C}_t^{-1} \left( t(\boldsymbol{z} - \boldsymbol{\mu}) + \sigma_t \boldsymbol{\epsilon} \right). \tag{55}$$

Applying Eq. 20 and taking the expectation over $\boldsymbol{\epsilon}$ yields

$$\ell_t^\star(\boldsymbol{z}) = c_t + \sigma_t^2 (\boldsymbol{z} - \boldsymbol{\mu})^\top \boldsymbol{C}_t^{-2}(\boldsymbol{z} - \boldsymbol{\mu}), \tag{56}$$

where $c_t$ is independent of $\boldsymbol{z}$. Therefore,

$$\mathcal{L}_{\text{align}}^\star(\boldsymbol{z}) = C_P + (\boldsymbol{z} - \boldsymbol{\mu})^\top \left( \int_0^1 \sigma_t^2 \boldsymbol{C}_t^{-2} \, \mathrm{d}t \right) (\boldsymbol{z} - \boldsymbol{\mu}), \tag{57}$$

for a constant $C_P$ independent of $\boldsymbol{z}$.

Let $\boldsymbol{\Sigma} = \boldsymbol{U} \operatorname{diag}(\lambda_1, \dots, \lambda_{d_1}) \boldsymbol{U}^\top$. Functional calculus reduces the matrix integral to the scalar identity

$$\int_0^1 \frac{(1-t)^2}{(t^2 \lambda + (1-t)^2)^2} \, \mathrm{d}t = \frac{\pi}{4\sqrt{\lambda}}, \qquad \lambda > 0. \tag{58}$$

To verify it, set $r = t/(1-t)$, so that $t = r/(1+r)$, $1 - t = 1/(1+r)$, and $\mathrm{d}t = \mathrm{d}r/(1+r)^2$. The integral becomes

$$\int_0^\infty \frac{\mathrm{d}r}{(1 + \lambda r^2)^2} = \frac{1}{\sqrt{\lambda}} \int_0^\infty \frac{\mathrm{d}u}{(1 + u^2)^2} = \frac{\pi}{4\sqrt{\lambda}}. \tag{59}$$

Hence,

$$\int_0^1 \sigma_t^2 \boldsymbol{C}_t^{-2} \, \mathrm{d}t = \frac{\pi}{4} \boldsymbol{\Sigma}^{-1/2}, \tag{60}$$

which proves Eq. 22.

If $\boldsymbol{\Sigma} = s^2 \boldsymbol{I}$, then

$$\mathcal{L}_{\text{align}}^\star(\boldsymbol{z}) = C_P + \frac{\pi}{4s} \|\boldsymbol{z} - \boldsymbol{\mu}\|^2. \tag{61}$$

Since

$$-\log p_{\text{prior}}(\boldsymbol{z}) = \frac{1}{2s^2} \|\boldsymbol{z} - \boldsymbol{\mu}\|^2 + \frac{d_1}{2} \log(2\pi s^2), \tag{62}$$

we obtain

$$\mathcal{L}_{\text{align}}^\star(\boldsymbol{z}) = \frac{\pi s}{2} \left[ -\log p_{\text{prior}}(\boldsymbol{z}) \right] + b_s \tag{63}$$

for a constant $b_s$, proving Eq. 23.

### A.5 Proof of Proposition 4

Let $\boldsymbol{Z} \sim Q$, where $Q$ has finite second moment, and let $\boldsymbol{Y} = \sigma_t \boldsymbol{\epsilon} + t\boldsymbol{Z}$. Write

$$\boldsymbol{S}_Z = \nabla_{\boldsymbol{y}} \log k_t^{\boldsymbol{Z}}(\boldsymbol{Y}), \tag{64}$$

$$\boldsymbol{S}_Q = \nabla_{\boldsymbol{y}} \log q_t(\boldsymbol{Y}), \tag{65}$$

$$\boldsymbol{S}_P = \nabla_{\boldsymbol{y}} \log p_t(\boldsymbol{Y}), \tag{66}$$

where $q_t$ is the density of $Q_t$. Differentiating the Gaussian-mixture representation of $q_t$ yields the conditional-score identity

$$\boldsymbol{S}_Q = \mathbb{E}[\boldsymbol{S}_Z \mid \boldsymbol{Y}]. \tag{67}$$

Therefore,

$$\mathbb{E}\left[\langle \boldsymbol{S}_Z - \boldsymbol{S}_Q, \boldsymbol{S}_Q - \boldsymbol{S}_P \rangle\right] = 0, \tag{68}$$

and the conditional-expectation Pythagorean identity gives

$$\mathbb{E}\left[\|\boldsymbol{S}_Z - \boldsymbol{S}_P\|^2\right] = \mathbb{E}\left[\|\boldsymbol{S}_Z - \boldsymbol{S}_Q\|^2\right] + \mathcal{I}(Q_t\|P_t). \tag{69}$$

The Gaussian-mixture scores satisfy

$$\boldsymbol{S}_Z = \frac{t\boldsymbol{Z} - \boldsymbol{Y}}{\sigma_t^2}, \tag{70}$$

$$\boldsymbol{S}_Q = \frac{t\mathbb{E}[\boldsymbol{Z} \mid \boldsymbol{Y}] - \boldsymbol{Y}}{\sigma_t^2}. \tag{71}$$

Hence,

$$\boldsymbol{S}_Z - \boldsymbol{S}_Q = \frac{t}{\sigma_t^2}\left(\boldsymbol{Z} - \mathbb{E}[\boldsymbol{Z} \mid \boldsymbol{Y}]\right), \tag{72}$$

and

$$\mathbb{E}\left[\|\boldsymbol{S}_Z - \boldsymbol{S}_Q\|^2\right] = \frac{t^2}{\sigma_t^4}\,\mathrm{mmse}_Q(t). \tag{73}$$

Multiplying the Pythagorean identity by $\sigma_t^2/t^2$ and applying Proposition 2 gives

$$\mathbb{E}_{\boldsymbol{Z}\sim Q}[\ell_t^\star(\boldsymbol{Z})] = \frac{\sigma_t^2}{t^2}\mathcal{I}(Q_t\|P_t) + \frac{\mathrm{mmse}_Q(t)}{\sigma_t^2}, \tag{74}$$

which proves Eq. 26. Integration over $t$ proves Eq. 27 whenever the integral is finite.

For completeness, suppose $\mathcal{I}(Q_t\|P_t) = 0$ at a fixed $t \in (0,1)$. Since $p_t$ and $q_t$ are smooth, positive, and normalized, equality of their scores almost everywhere implies $q_t = p_t$. Their characteristic functions satisfy

$$\widehat{Q_t}(\boldsymbol{\omega}) = \widehat{Q}(t\boldsymbol{\omega}) \exp\left(-\tfrac{1}{2}\sigma_t^2\|\boldsymbol{\omega}\|^2\right), \tag{75}$$

$$\widehat{P_t}(\boldsymbol{\omega}) = \widehat{P}(t\boldsymbol{\omega}) \exp\left(-\tfrac{1}{2}\sigma_t^2\|\boldsymbol{\omega}\|^2\right). \tag{76}$$

The Gaussian factor is nonzero, so $Q_t = P_t$ implies $\widehat{Q}(t\boldsymbol{\omega}) = \widehat{P}(t\boldsymbol{\omega})$ for every $\boldsymbol{\omega}$, and therefore $Q = P$.

## A.6 Proof of Proposition 5

On the joint probability space of $(t, \boldsymbol{Z}, \boldsymbol{\epsilon})$, define

$$\boldsymbol{A} = \boldsymbol{v}_\theta(\sigma_t\boldsymbol{\epsilon} + t\boldsymbol{Z}, t) - (\boldsymbol{Z} - \boldsymbol{\epsilon}), \tag{77}$$

$$\boldsymbol{B} = \boldsymbol{v}_P^\star(\sigma_t\boldsymbol{\epsilon} + t\boldsymbol{Z}, t) - (\boldsymbol{Z} - \boldsymbol{\epsilon}). \tag{78}$$

Then

$$\|\boldsymbol{A}\|_{L^2} = \sqrt{J_\theta(Q)}, \qquad \|\boldsymbol{B}\|_{L^2} = \sqrt{J^\star(Q)}, \tag{79}$$

and

$$\|\boldsymbol{A} - \boldsymbol{B}\|_{L^2} = \sqrt{\Delta_\theta(Q)}. \tag{80}$$

The reverse triangle inequality in the Hilbert space $L^2$ gives

$$\left|\|\boldsymbol{A}\|_{L^2} - \|\boldsymbol{B}\|_{L^2}\right| \leq \|\boldsymbol{A} - \boldsymbol{B}\|_{L^2}, \tag{81}$$

which is Eq. 31.

## B  Additional Toy Details

Fig. 5 provides the complete visualization for the additional two-dimensional prior distributions used in Sec. 5.1: a Grid of Gaussians, Two Moons, Concentric Rings, a Spiral, and a Swiss Roll. For each distribution, following the visualization style of Fig. 3, we illustrate: (a) The optimized variables $\boldsymbol{z}$ (red triangles) and samples from $p_{\text{prior}}$ (blue dots), overlaid on the analytic or KDE-estimated negative log-likelihood (NLL) landscape of $p_{\text{prior}}$. (b) The landscape of the alignment loss $\mathcal{L}_{\text{align}}$ (background heatmap), with samples from $p_{\text{prior}}$ (blue dots). (c) The evolution of $\mathcal{L}_{\text{align}}(\boldsymbol{z}; \theta)$ (blue solid line) and the analytic or KDE-estimated NLL during the optimization of $\boldsymbol{z}$.

For the Grid of Gaussians, which is also a mixture of Gaussians, the exact NLL $-\log p_{\text{prior}}(\boldsymbol{z})$ is computed analytically. For the other distributions (Two Moons, Concentric Rings, Spiral, and Swiss Roll), where an analytical form for $p_{\text{prior}}$ is not readily available, we estimate the NLL using Kernel Density Estimation (KDE). This estimation is based on $N = 100,000$ samples drawn from the respective $p_{\text{prior}}$ and employs a Gaussian kernel with a bandwidth of $h = 0.1$. The probability density $\hat{p}_{\text{prior}}(\mathbf{x})$ at a point $\mathbf{x}$ is estimated as:

$$\hat{p}_{\text{prior}}(\mathbf{x}) = \frac{1}{Nh^d} \sum_{i=1}^{N} K\left(\frac{\mathbf{x} - \mathbf{x}_i}{h}\right), \tag{82}$$

where $\mathbf{x}_i$ are the $N$ samples drawn from $p_{\text{prior}}$, $d$ is the dimensionality (here, $d = 2$), and $K(\cdot)$ is the Gaussian kernel function. The NLL for an optimized variable $\boldsymbol{z}$ is then approximated by $-\log(\hat{p}_{\text{prior}}(\boldsymbol{z}))$. This provides an empirical measure of how well $\boldsymbol{z}$ aligns with the prior distribution as estimated by KDE.

The complete landscape comparison is shown in Fig. 5. The optimization trajectories are shown in Fig. 6. Together, these visualizations show that minimizing $\mathcal{L}_{\text{align}}$ consistently moves the optimized variables toward high-density regions and that the alignment loss closely follows the analytic or KDE-estimated NLL trend across the toy priors.

## C  Additional Implementation Details

### C.1  Implementation Details of the Toy Example

The primary toy example, illustrated in Figure 3, utilizes a 2D Mixture of Gaussians (MoG) as the prior distribution $p_{\text{prior}}(\boldsymbol{x})$. This MoG distribution consists of 5 components, each with an isotropic standard deviation of 0.3. The means of these Gaussian components are distributed evenly on a circle of radius 3.0. Prior to model training, samples drawn from this MoG distribution are normalized by dividing by their standard deviation, which is empirically computed from a large batch of 10 million samples. The same flow-model and latent-optimization setup is used for the additional toy distributions in Appendix B.

The conditional flow model, denoted $\boldsymbol{v}_\theta(\boldsymbol{x}, t)$, is implemented using a MLP with AdaLN. This network has 2 input channels, 2 output channels, a hidden dimensionality of 512, and incorporates 4 residual blocks. The flow model is trained for $1m$ steps using the AdamW optimizer (beta values of (0.9, 0.999) and no weight decay) with a constant learning rate of $1 \times 10^{-4}$, and a batch size of 256.

A set of 1,000 learnable latent variables $\{\boldsymbol{z}_i\}$ is initialized by sampling from a standard normal distribution $\mathcal{N}(\mathbf{0}, \boldsymbol{I})$. These latents are then optimized to align with the prior distribution $p_{\text{prior}}$ by minimizing the alignment loss $\mathcal{L}_{\text{align}}$. This alignment training phase also employs the Adam optimizer (betas=(0.9, 0.999), no weight decay), with a learning rate of $1 \times 10^{-2}$, and runs for 5,000 steps.

### C.2  Implementation Details of the Flow Model

The flow model $\boldsymbol{v}_\theta(\boldsymbol{z}, t) : \mathbb{R}^{d_1} \times [0, 1] \to \mathbb{R}^{d_1}$ is implemented as a multi-layer perceptron (MLP) with 6 layers and 1024 hidden units per layer. The network employs GELU activation functions and incorporates time modulation through adaptive layer normalization (AdaLN) to handle the temporal dimension $t$. When dimension mismatch occurs between the latent space dimension $d_1$ and prior feature space dimension $d_2$,

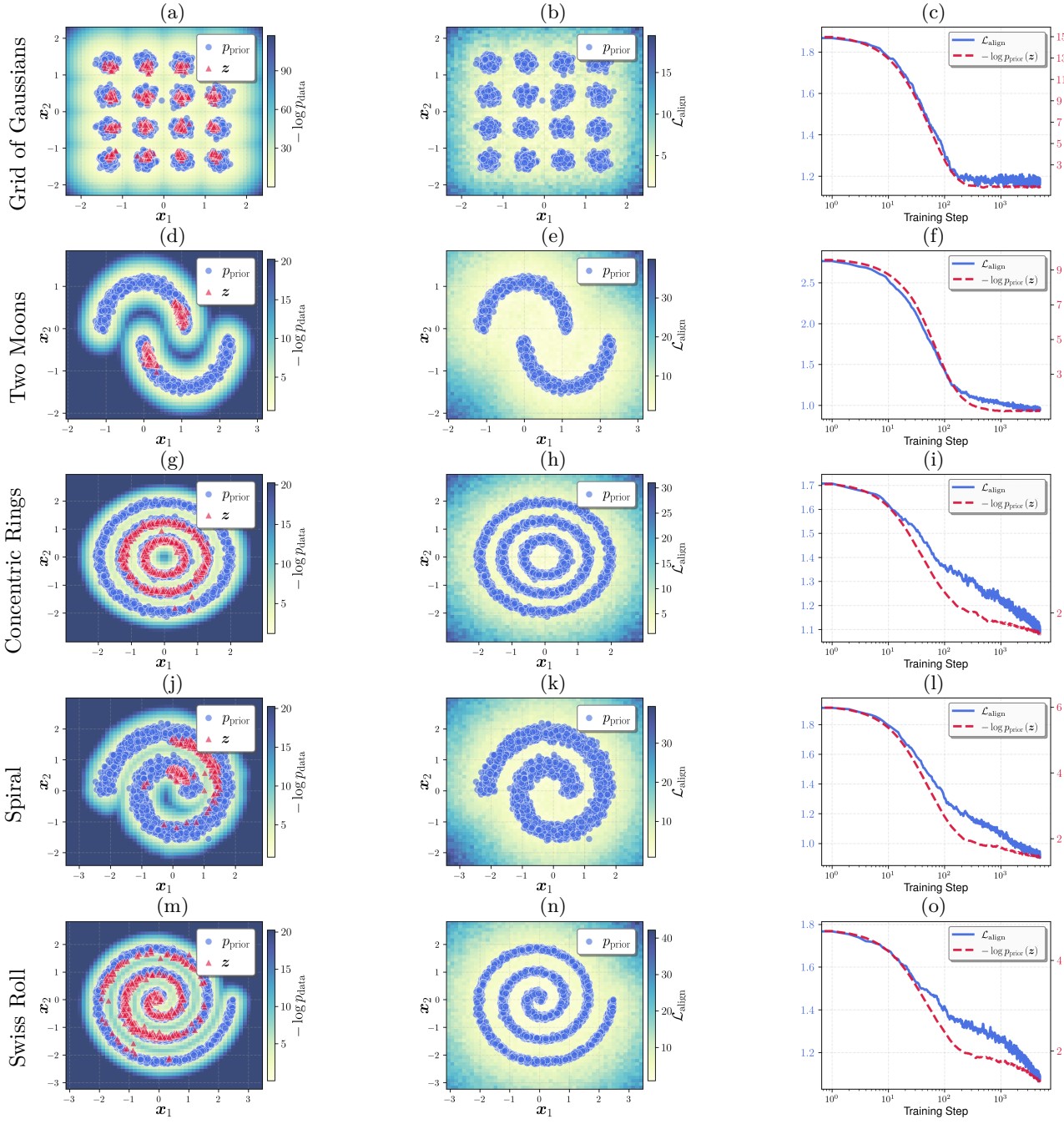

Figure 5: Additional two-dimensional priors. In each row, the left panel shows optimized variables $\boldsymbol{z}$ and samples from $p_{\mathrm{prior}}$ over an analytic or KDE-estimated NLL landscape, the middle panel shows the alignment-loss landscape, and the right panel tracks $\mathcal{L}_{\mathrm{align}}(\boldsymbol{z};\theta)$ together with the corresponding NLL during optimization. Across all priors, the optimized variables concentrate in high-density regions and the alignment loss closely follows the NLL trend.

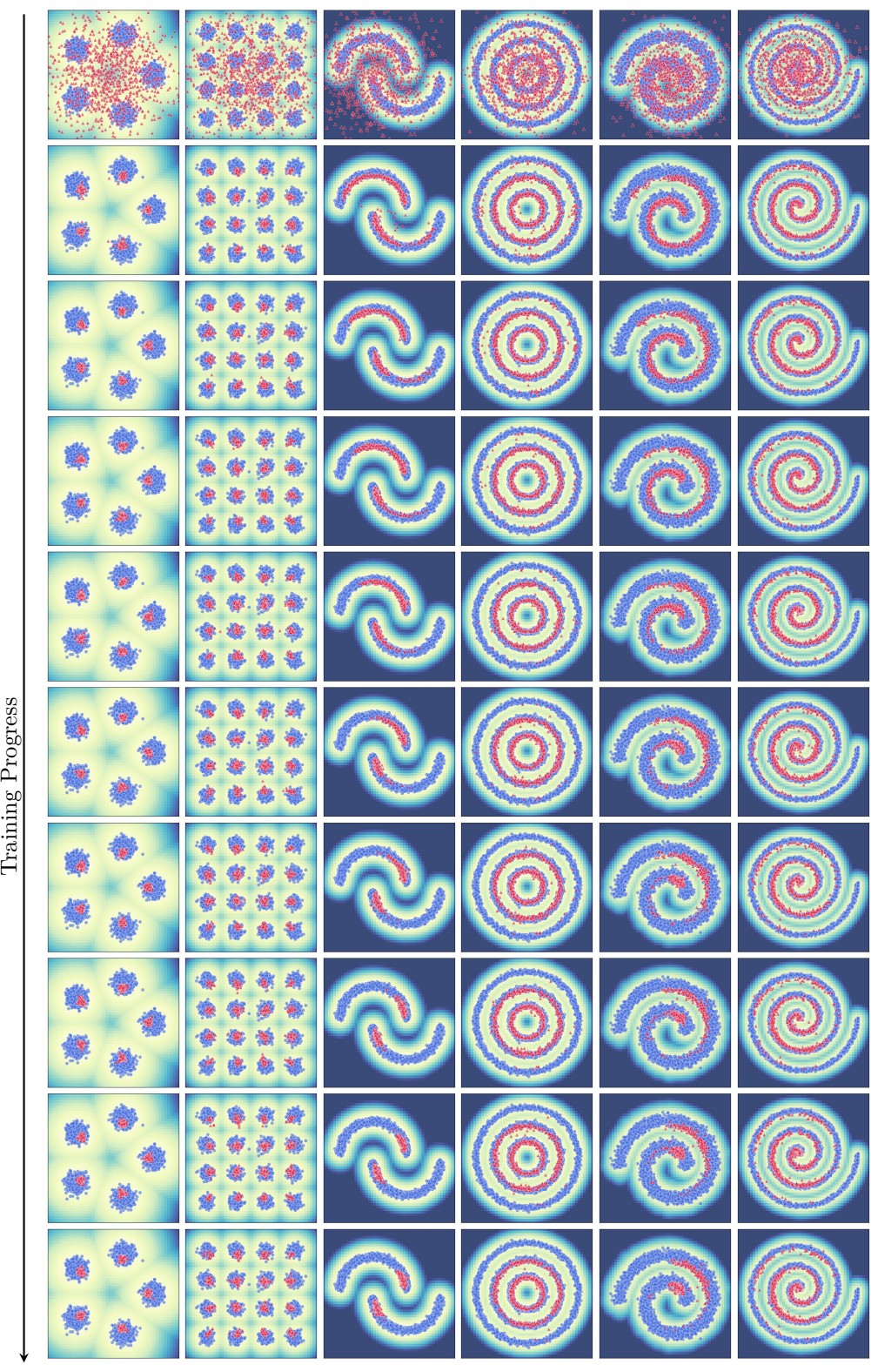

Figure 6: Evolution of the optimized variables $z$ (red triangles) during training across the toy priors. Each column represents one prior distribution $p_{\text{prior}}$, with optimization progressing vertically. Minimizing $\mathcal{L}_{\text{align}}$ consistently guides $z$ toward low-NLL, high-density regions under the analytic or KDE-estimated NLL landscape.

fixed linear projection layers are applied to map prior features to the appropriate dimension. These projection matrices are initialized with Gaussian weights scaled by $1/\sqrt{d_2}$ and remain frozen during training.

The flow model is trained using the flow matching objective on the prior distribution $p_{\text{prior}}$ for 1 million steps. During training, the model learns the conditional mean velocity on straight interpolants between a standard Gaussian base sample and an independent sample from the prior feature distribution. The resulting field captures the Gaussian-smoothed prior geometry across interpolation times and is frozen for subsequent latent-space alignment. For continuous prior distributions, the same field can also be used to define an approximate generative ODE; for atomic priors such as VQ codebook embeddings, the alignment objective operates on the Gaussian-smoothed intermediate distributions. Detailed hyperparameters are provided in Table 3.

Table 3: Training hyperparameters used for the flow prior, autoencoder, and MAR-B generator.

| Hyperparameter | Flow | Autoencoder | MAR |
|---|---|---|---|
| Global Batch Size | | 256 | |
| Steps | $1m$ | $50k$ | $250k$ |
| Optimizer | | AdamW | |
| Base Learning Rate | | $1.0 \times 10^{-4}$ | |
| LR Scheduler | Cosine | Cosine | Constant |
| Warmup Steps | 2.5k | 2.5k | 62.5k |
| Adam $\beta_1$ | | 0.9 | |
| Adam $\beta_2$ | 0.95 | 0.95 | 0.999 |
| Weight Decay | $1.0 \times 10^{-4}$ | $1.0 \times 10^{-4}$ | 0.02 |
| Max Grad Norm | | 1.0 | |
| Mixed Precision | | BF16 | |
| EMA Rate | | 0.9999 | |

## C.3 Implementation Details of Autoencoders

Our autoencoder architecture follows the SoftVQ design, which employs Vision Transformer (ViT) based encoder and decoder networks. The encoder utilizes a ViT-Large model with patch size 14 from DINOv2 (Oquab et al., 2024), initialized with pre-trained weights and fine-tuned with full parameter updates during training. The decoder employs the same ViT-Large architecture but is initialized randomly without pre-trained weights.

The training process utilizes adversarial loss with a DINOv2-based discriminator, incorporating patch-based adversarial training with hinge loss formulation. Perceptual loss is applied using VGG features with a warmup period of $10k$ steps. The model is trained for $50k$ steps with cosine learning rate scheduling and exponential moving averages for stable training dynamics. Unlike SoftVQ, we do not employ the sample-level alignment loss (i.e., REPA loss), making our method more general and efficient. Detailed hyperparameters are provided in Table 3.

We followed the SoftVQ implementation as closely as possible. While we can reproduce almost identical reconstruction results, our tokenizer doesn't quite match the generation performance of the released pre-trained model, even after significant effort to optimize it. We believe this gap comes from differences in the cleaned-up code and the specific hardware we used for training. To keep things fair and validate the effectiveness of our method, we conduct all experiments on *the same hardware with identical training settings*.

## C.4 No-Alignment Diagnostic

To verify that the ImageNet alignment trend is induced by the proposed objective, we also train an autoencoder with the same Dino prior target but remove the alignment loss. As shown in Fig. 7, the $k$-NN distance to the prior does not improve in this setting.

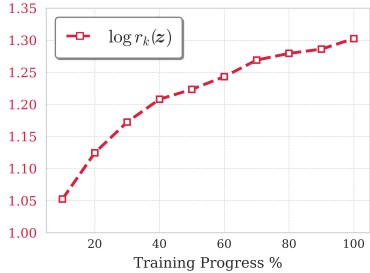

Figure 7: Alignment behavior without the proposed loss. The $k$-NN distance metric $\log r_k(\boldsymbol{z})$ does not improve for an autoencoder trained with the same Dino prior target but without the alignment objective.

## C.5    Implementation Details of MAR

We follow the original MAR-B implementation with several key modifications. We incorporate qk-norm in the attention mechanism and replace the diffusion head with a flow-based head trained using per-token flow matching loss. The original SD-KL-16 autoencoder is replaced with our trained autoencoders, applying input normalization with scaling factor 1.7052 estimated from sample batches.

Our model uses MAR-B architecture with $256 \times 256$ input images. The flow-based MLP head features adaptive layer normalization with 6 layers and 1024 hidden units per layer, identical to the original diffusion implementation. The model processes sequences of length 64 corresponding to our 64-token latent representation. More training details are provided in Table 3.

For inference, we employ an Euler sampler with 100 steps for the flow-based generation. The autoregressive sampling is limited to 64 steps. Generation uses batch size 256 and produces 50,000 images for evaluation. All evaluations use the standard toolkit from guided diffusion with FID and IS metrics computed at regular intervals during training.

## D    Ethics Statement

Since our framework advances the capabilities of generative image synthesis, it shares the broader societal implications associated with deep generative models, such as the potential for misuse in creating misleading content, deepfakes, or copyright infringement. Furthermore, as our method aligns latent spaces to arbitrary pre-trained priors, there is a risk of propagating or amplifying biases inherent in those underlying foundation models.

