# OpenReview forum: "Aligning Latent Spaces with Flow Priors"
_TMLR — Under review for TMLR_

### Review · Reviewer_qb7t · 2026-07-15

**Summary Of Contributions:**

This paper proposes to align latent spaces to a pretrained flow model, based on the well-motivated core idea that structured latents are easier for downstream generative modeling. The method answers the question: can a learnable latent space be aligned to an arbitrary sample-defined prior using a frozen flow model?
The approach consists of two stages: (1) pretraining a flow model on the target prior distribution, and (2) using this frozen flow as a regularizer via an alignment loss that avoids expensive ODE solving and Jacobian-trace computation.

Key Strengths:
- Well-motivated problem and lightweight computational approach once the flow is pretrained
- Theoretical analysis showing alignment loss relates to likelihood for Gaussian priors
- Empirical validation on toy examples and large-scale ImageNet experiments

Key Weaknesses:
- Evaluation limited to a single generative model (MAR-B)
- Prior selection remains unsolved and probably task-dependent; no principled guidance
- Claims about interpretability and domain-specific knowledge in the introduction are not validated

**Additional Comments:**

Minor issues:

- Eq. 4: This should be labeled as the Conditional Flow Matching (CFM) objective, not FM
- Fig. 2(b) caption: The blue arrow for "steers $y_{\mathrm{bad}}$ towards low-energy regions" is difficult to see with the current color scheme (it looks dark green); use more distinguishable colors please

**Audience:**

Yes

**Audience Explanation:**

The paper's findings are of clear interest to the representation learning and generative modeling community. The flow-based alignment approach is interesting and supported by the theoretical analysis.

However, the practical utility remains unclear. The paper does not solve the prior-selection problem, i.e., which prior to use remains task-dependent and unclear. That the VQ prior fails while simple textual features match sophisticated visual features is really interesting but unexplained. Without further guidance on this, the method is presented rather as an interesting "idea". This currently limits its immediate adoption and impact.

**Broader Impact Concerns:**

No concerns on the ethical implications apart from overall concerns that apply to the whole field of generative modeling.

**Claims And Evidence:**

Yes

**Claims Explanation:**

The core theoretical claims are rigorously supported: Proposition 2 establishes that the alignment loss is an exact multiscale Fisher energy for any prior; Proposition 3 proves exact NLL equivalence for isotropic Gaussians. The empirical claims are well-validated: alignment loss closely follows with true NLL in toy experiments, correlates with k-NN density proxies on ImageNet, and improves MAR-based generation.

However, the paper's broader claims (that alignment enhances interpretability, incorporates domain knowledge, and benefits generative modeling generally) are not fully supported. These claims are only partially supported through MAR evaluation on ImageNet. Validation on additional generative models and non-generative tasks would strengthen the evidence for these broader assertions.

**Requested Changes:**

- Evaluate on at least one additional generative model. Currently, alignment is shown beneficial only for MAR-B. Adding e.g. a latent diffusion model would validate whether alignment benefits generative modeling more broadly or is specific to autoregressive models. This is essential to support the general claim that structured latents are easier for downstream generative modeling.
- Evaluate on non-generative downstream tasks. The introduction claims alignment enhances interpretability and facilitates domain-specific learning, but only generation is tested. Adding a downstream classification task (or interpretability?) would help to further clarify the benefits of structured latents.
- Analyze prior-selection failure. Discuss why the VQ prior fails while textual Qwen features succeed. Is it a hyperparameter issue ($\lambda$ tuning)? A fundamental limitation of discrete priors?
- (If feasible) Include comparison to DMVAE. This is the closest existing baseline for arbitrary prior matching. Without a head-to-head comparison, it is unclear whether the flow-based approach offers advantages over this competing method. This comparison should be added to Table 1 or discussed in detail.

---

### Review · Reviewer_D9Zx · 2026-07-16

**Summary Of Contributions:**

The paper addresses the problem of aligning an autoencoder's latent representations with a data-driven prior distribution. In conventional autoencoder settings, latent variables are typically regularized toward a simple prior, such as a Gaussian. This choice may be overly restrictive or inefficient when the true latent structure we aim to obtain is more complex, and the authors propose aligning the latent representations with a learned prior distribution parametrized by a flow-matching model.

---
### Strengths

The core idea is intuitive and practical. The proposed alignment objective produces gradients that encourage the latent codes to conform to the learned prior by constraining noisy versions of a latent code, sampled along the flow-matching trajectory, to be accurately mapped back to the clean latent representation using the trained prior model.

The authors also provide a theoretical interpretation of this objective, showing that it can be viewed as a multiscale prior energy regularization, where low energies indicate latent samples lying in high-density regions, as learned by the prior model.

To illustrate the method, the paper first presents a toy experiment in which samples are inferred directly from a learned prior by minimizing the alignment objective. The resulting samples cluster around the modes of the prior distribution, providing an intuitive demonstration of how the proposed mechanism pushes latents to the target region.

Then, in a more practical setting, the authors train an autoencoder whose latent space is aligned with different pre-trained representations. They evaluate the resulting model in terms of both reconstruction quality and its usefulness as an intermediate representation for training a generative model. Their results show that aligning the autoencoder latents with a learned latent prior has the potential to improve both reconstruction fidelity and generation quality.

The main weakness is the overreliance on improving reconstruction and generation quality as the main result of the paper.
The numbers shown in the main ImageNet experiment, show that the alignment in some cases improves reconstruction or generation but this improvement is never dramatical over the base, KL-regularized autoencoder.
The authors should have tried to connect the resulting autoencoder latent more with the representations used in constraining the model.
For instance, investigating different behaviors of the model when aligning with a text prior (Qwen) vs. a semantic visual prior (DINOv2).

---
### Weaknesses

The main weakness of the paper is its heavy reliance on improving reconstruction and generation quality as the primary empirical contribution. In the main ImageNet experiment, the proposed alignment method only sporadically improves reconstruction/generation, and the gains over the baseline KL-regularized autoencoder are not substantial.

The evaluation would be more compelling if the authors had more directly investigated how the properties of the learned autoencoder's latent space depended on the representation used for alignment. For example, there should be a meaningful difference other than FID/PSNR between models aligned with a text-based prior (Qwen) versus those aligned with a semantic visual prior (DINOv2). By only probing the downstream capabilities of the autoencoder, such differences are hard to decode, and overall, it fails to highlight the importance of choosing a prior distribution to align with.

---

This is a promising approach that begs for further experiments that better explain the impact of the choice of prior on the learned autoencoder's latent space.

**Additional Comments:**

- An interesting question for the authors to address: Why do you constrain the prior model to be a simple MLP? Could you instead train the prior with a small CNN (in terms of receptive field) that better captures spatial structures in the latent space? At what point does the prior become too complex, and the autoencoder becomes an unnecessary step before the generator?

**Audience:**

Yes

**Audience Explanation:**

The findings presented are interesting for the wider TMLR audience that works on autoencoders and generative models. There is a lot of discussion around choosing the right latent space for training a generative model in, and the paper introduces a new method to construct such latent spaces.

**Broader Impact Concerns:**

There are no major ethical implications of this work. An ethics statement is provided in Appendix D.

**Claims And Evidence:**

Yes

**Claims Explanation:**

The main claim of the paper is that it is possible to align with a data-driven prior using the proposed alignment objective and prior flow-matching model. Both the theoretical framework discussed and the toy experiment show that the learned autoencoder's latent space ends up being aligned with the chosen prior.

**Requested Changes:**

- The paper would greatly benefit from a more detailed comparison of the latent spaces induced by the different priors used. For example, the authors could test whether autoencoder latents aligned with DINOv2 can be clustered into semantic regions better than latents aligned with the low-level VAE features, since DINOv2 is known to carry more semantic information. Similarly, can Qwen-aligned latents be more easily decoded into text captions? Running such lightweight experiments on autoencoder latents obtained from real/generated images would provide a more detailed comparison than relying on marginal FID and PSNR improvements.

- The authors use the same MLP for all prior models. However, the complexity of each space is vastly different (both in dimensionality and semantics), which raises the question of whether the comparisons made between different priors are fair. A simple ablation showing that the complexity of the MLP saturates the results obtained would help determine this.

- The MAR-B generator is not properly introduced in the paper. For a reader who is not familiar with the latest generative models, there is a significant jump from prior --> autoencoder --> generator in Section 5.2. I suggest that the authors first explain why they chose to evaluate both reconstruction and generation (latent diffusion models, etc.), and then introduce the scheme they chose for their experiment.

---

### Review · Reviewer_32YU · 2026-07-17

**Summary Of Contributions:**

**Summary**

The paper introduces a methodology for aligning the latent space (of some latent model) with some sample-defined prior distribution. Initially, a flow matching model is trained to learn the mapping between a base distribution to some sample-defined feature space. Then, the learned flow is used as a target prior for a learnable latent model. The latent-to-prior alignment is realized through minimizing a theoretically-grounded and tractable alignment objective. Initial experiments conducted in 2D synthetic and high-resolution image data (i) demonstrated that the proposed methodology is an effective proxy for aligning learned latent space into sample-defined prior. Finally, experiments on high-resolution image data (ii) demonstrated the impact of the proposed method in image reconstruction and generation.

**Strengths**

S1. I found the idea of aligning arbitrary latent spaces via flow prior interesting.

S2. The experimental analysis is intuitively constructed where it first grounds understanding through sanity checks (e.g., alignment objective being a proxy of the nnl minimization etc.) prior to moving to discussing the implication of the method (e.g., image generation setting).

S3. The proposed method appears to be grounded/supported by theory.

**Weaknesses**

W1. Some parts of the paper are difficult to parse. Crucially, I found the theoretical grounding/support section 4.3. difficult to follow.

W2a. I found the experimental results to not be convincing regarding the effectiveness of the proposed method. Although interesting ablations were conducted, these do not hint towards a clear narrative and conclusive takeaways.

W2b. Each experiment appears to have been conducted only once. If this is the case, it is impossible to tell whether the difference in performance is driven by the methods used or by random variation.

W3. The paper claims the proposed method is lightweight but concrete evidence was not provided. That is, how does the method being lightweight translate in practice in relation to the baseline alternatives.

**Audience:**

Yes

**Audience Explanation:**

The paper proposes a novel method for latent space alignment while also developing theoretical arguments to support it. Despite the empirical evidence being mixed, in my view the work suffice for concept feasibility status. Based on these, one could reasonable assume that either the theoretical or the methodological contribution could be of interest for part of the tmlr community.

**Broader Impact Concerns:**

None.

**Claims And Evidence:**

No

**Claims Explanation:**

- Effectiveness/improvement over baseline claims: When looking at Tab. 1., the results appear to be mixed. That is, I am not convinced that one can claim the proposed method is effective or that it improves over the baselines in image generation.

For example, the best performing, for generation, flow prior (semantic) improves FID/IS at the cost of reconstruction, precision and recall. For a method to be effective with respect to some target quality, one would expect to improve the said target quality without sacrificing too much of some other equally important quality such as reconstruction. Otherwise, the method would better be characterized as inducing a trade-off between some relevant qualities.

Additionally, it is unclear how the picture would change if the experiments were to be reported across multiple independent runs. As discussed earlier, one can not be certain for the extend to which the differences in the reported metrics are driven by the methods or random variation.

- Claims on the framework being lightweight: In this regard, the claims were not substantiated as time and/or computational resource comparisons were not provided.

**Requested Changes:**

(Critical) RC1. Could it be possible to report the mean and variance for each metric considered in the image generation experiment? Doing so will factor out the effect of random variation when comparing the proposed method with the baselines.

(Critical) RC2. Do the results across multiple runs support the effectiveness claim made for the proposed method?

(Critical) RC3. Could you provide some quantitative analysis on how the framework being lightweight translates in practice? Although one would intuitively expect this to be the case, it would be interesting to see how the proposed method compares to the baselines in this regard.

(Critical) RC4. Could you provide a road map in the beginning of section 4.3. briefly summarizing the motivation behind introducing proposition 1-5 as well as how these are connected?

(Critical) RC5. Is there any way to disambiguate the impact between having a more flexible prior and utilizing the latent structure from a pretrained model? The current experimental setting entangles these two aspects. For example one could consider reporting the performance of KL + MoG learnable prior (instead of a standard gaussian).

(Critical) RC6. Could you please clarify how the proposed method relates to simply fitting a parametric prior (e.g., MoG prior) into the latent space of some pretrained model (e.g., dino) and consequently regularizing the latent space to match that learned parametric prior?

(Strengthen) RC7. In Sec 5.1., the samples appear to collapse to parts of the prior distribution (even more evident in other distributions provided in the appendix), namely, the optimized latent does not cover the full prior distribution. Such behavior is a byproduct of only fitting the latent to the prior without having any objective to encourage spreading (e.g., reconstruction). This does not interfere with the goal of the section which is to show that the alignment objective is a proxy to the intractable nll. If my understanding is correct, I suggest you briefly clarify with respect to these aspects, if not could you please elaborate on whether the mode collapse behavior is observed and how this affects the claims?

(Strengthen) RC8. What is the motivation behind conducting the ablations in Table 2 for the textual flow-prior? Also, the first row of Table 2a matches the last row of Table 1, second to last row of Table 2b also match the last row of Table 1?

(Strengthen) RC9. Could you elaborate how the NN-based nll proxy was constructed for the image data? I believe that providing a visual illustration could be of help in this regard.

(Minor) RC10. Fig 2., labels horizontal and vertical axes as x1 and x2 respectively which could potentially be confused with the flow prior notation x1 used earlier. If the axes are not connected to the flow prior notation, consider changing their name. If these are connected, could you please clarify their connection?